# Research

ecology, evolution, genetics

*Crassostrea virginica*, heritability, quantitative genetics, rapid evolution, transgenerational plasticity

**Author for correspondence:**
Joanna S. Griffiths
e-mail: jsgriffiths@ucdavis.edu

One contribution to a Special Feature: Evolution in Changing Seas. Guest edited by Katie E. Lotterhos, Molly Albecker and Geoffrey Trussell.

# Transgenerational plasticity and the capacity to adapt to low salinity in the eastern oyster, *Crassostrea virginica*

Joanna S. Griffiths[1], Kevin M. Johnson[2], Kyle A. Sirovy[3], Mark S. Yeats[3], Francis T. C. Pan[5], Jerome F. La Peyre[4] and Morgan W. Kelly[3]

[1]Department of Environmental Toxicology and Department of Wildlife, Fish, and Conservation Biology, University of California, 4121 Meyer Hall, Davis, CA 95616, USA
[2]Biological Sciences Department, California Polytechnic State University, San Luis Obispo, CA, USA
[3]Department of Biological Sciences, and
[4]Department of Veterinary Science, Louisiana State University, Baton Rouge, LA, USA
[5]Department of Biological Sciences, University of Southern California, Los Angeles, CA, USA

JSG, 0000-0003-0319-515X; KMJ, 0000-0003-3278-3508; KAS, 0000-0003-3789-0257; FTCP, 0000-0002-1550-4581; MWK, 0000-0001-6998-5053

Salinity conditions in oyster breeding grounds in the Gulf of Mexico are expected to drastically change due to increased precipitation from climate change and anthropogenic changes to local hydrology. We determined the capacity of the eastern oyster, *Crassostrea virginica*, to adapt via standing genetic variation or acclimate through transgenerational plasticity (TGP). We outplanted oysters to either a low- or medium-salinity site in Louisiana for 2 years. We then crossed adult parents using a North Carolina II breeding design, and measured body size and survival of larvae 5 dpf raised under low or ambient salinity. We found that TGP is unlikely to significantly contribute to low-salinity tolerance since we did not observe increased growth or survival in offspring reared in low salinity when their parents were also acclimated at a low-salinity site. However, we detected genetic variation for body size, with an estimated heritability of 0.68 ± 0.25 (95% CI). This suggests there is ample genetic variation for this trait to evolve, and that evolutionary adaptation is a possible mechanism through which oysters will persist with future declines in salinity. The results of this experiment provide valuable insights into successfully breeding low-salinity tolerance in this commercially important species.

## 1. Introduction

Our rapidly changing climate will expose organisms to novel and potentially stressful environments. Many studies have demonstrated that species can adapt over ecological time scales that may keep pace with climate change [1], such as *Drosophila* in response to extreme heat waves [2] and *Brassica* plants in response to drought stress [3]. For evolution to occur, there must be heritable genetic variation present for traits that improve fitness in the novel environment [4]. Quantitative genetic studies test for the presence of genetic variation in a particular trait under selection and estimate the population's capacity to adapt to a new environment. If there is limited genetic variation within the population for tolerance to these emerging stressors, we would predict that there will be limited potential adaptation. These questions have led to a number of studies that have identified differing responses to emerging stressors between genotypes across a broad range of taxa, including plants, sea urchins, mussels, corals and fish [1,5–10].

Populations with low genetic variation may still be able to rely on phenotypic plasticity to promote survival in stressful and novel environments. Adaptive plasticity can move individuals closer to the physiological optimum for a novel environment, so that individuals that are more plastic have a higher fitness. However, plasticity can also be maladaptive, causing individuals

*Proc. R. Soc. B* **288**: 20203118

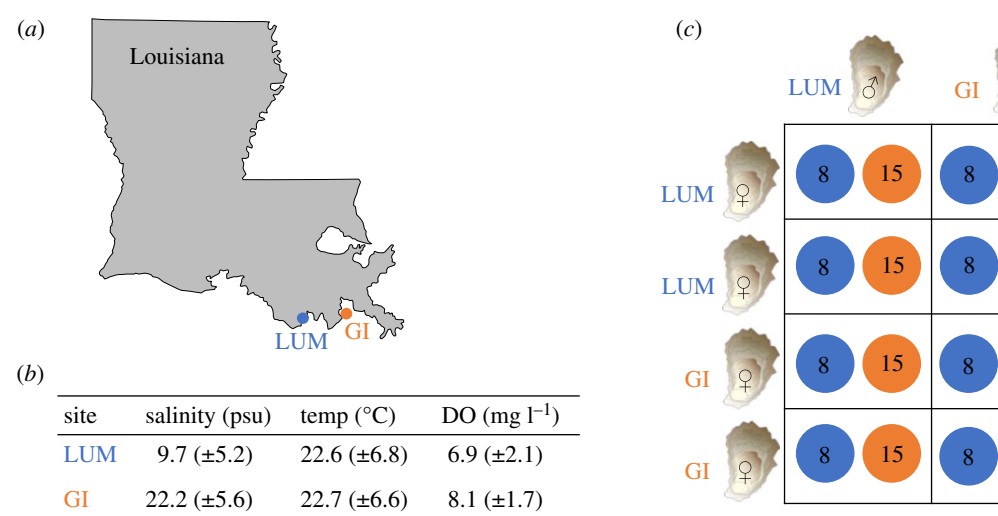

**Figure 1.** (*a*) Map of parental acclimation sites (LUM: LUMCON; GI: Grand Isle). (*b*) Mean daily water quality data for acclimation sites, with standard deviation in parentheses, during the acclimation period from 2017 to 2019 for LUM and GI. (*c*) Schematic of our modified North Carolina II block design (16 total blocks). For each experimental block, eggs from two individual dams from LUM were crossed with sperm from two individual sires, one from LUM and one from GI. Eggs from two individual dams from GI were crossed with sperm from two individual sires, one from LUM and one from GI. Offspring were reared at either ambient (15 psu) or low (8 psu) salinity treatments. Blue colours represent low salinities and orange represents ambient salinity. (Online version in colour.)

| site | salinity (psu) | temp (°C) | DO (mg l⁻¹) |
|------|----------------|-----------|-------------|
| LUM | 9.7 (±5.2) | 22.6 (±6.8) | 6.9 (±2.1) |
| GI | 22.2 (±5.6) | 22.7 (±6.6) | 8.1 (±1.7) |

to have a lower fitness in the novel environment [11]. Plasticity can be inherited across generations, termed transgenerational plasticity (TGP), whereby a plastic change induced in the parental population is inherited and expressed in the offspring generation [12–16].

Studies that attempt to identify mechanisms of TGP have traditionally focused on maternal investment, such as through mRNA, lipids or proteins passed on in the egg [17]. For example, egg quality has been demonstrated to be important for ocean acidification resistance in abalone [18]. Paternal effects have also been observed to affect offspring fitness, suggesting that sires contribute more than just genetic material, such as through epigenetic processes (e.g. DNA methylation and histone modifications) [19,20]. In cases where the parental environment accurately predicts and matches the offspring environment and the plastic response is adaptive, offspring inherit the parents' plastic trait change, thereby increasing fitness [21]. For example, when offspring of the marine tubeworm, *Galeolaria caespitosa*, are exposed to ecologically relevant temperatures, they have higher survival when the rearing temperature matches that of the parental environment [22]. Parental exposure to ecologically relevant stressful environments can also promote increased fitness in novel environments predicted under future climate change scenarios. Increased resistance to future low pH conditions was observed in *Strongylocentrotus purpuratus* sea urchins when their parents were conditioned in naturally low pH environments [23]. However, these beneficial effects may not always be prominent [24] and TGP may even act as a conduit for decreased fitness, transmitting harmful phenotypic effects to the next generation [11,25,26].

The interplay of intra-generational and TGP and evolution needs further investigation to accurately predict a species' response to climate change [27]. These mechanisms may work in tandem, whereby plasticity prevents a population from going extinct by extending the time over which selection has the opportunity to act [28]. The plastic response may even influence the evolutionary trajectory of the population by either speeding up or impeding the effects of selection [29]. However, TGP may not be enough to buffer populations from continually changing conditions [30];

therefore, the presence of additive genetic variation will play an important role in determining a population's response to future climate change.

To evaluate the capacity for adaptation to changing environments, we explore the influence of genetic variation and TGP on larval growth rates in the eastern oyster, *Crassostrea virginica*, an ecologically and economically important species in the northern Gulf of Mexico. *C. virginica* is a widespread euryhaline species but is sensitive to low salinities, with documented reductions in growth and production when salinity is below 15 psu [31]. While oysters are capable of withstanding extreme low salinity (less than 5 psu) for short periods of time, their tolerance rapidly declines in conjunction with high temperatures [32,33]. Under future climate change scenarios, precipitation is expected to increase in this region during peak temperatures in the summer and fall [34,35], drastically reducing the salinity in estuaries that oysters currently inhabit. In addition to climate change-related stressors, oysters will be more immediately affected by anthropogenic alterations to local hydrology. Coastal protection plans in this region include large-scale (approx. 7000 m³ s⁻¹) diversions of the Mississippi river, which will dramatically increase freshwater flow into currently productive oyster habitats [36].

In this study, we investigated the influence of the environment, TGP and additive genetic variation on fitness-related traits in *C. virginica* exposed to stressful low salinities. We first investigated the effects of the parental environment by rearing hatchery-bred oysters at either a low- or medium-salinity field site for 2 years (figure 1*a*) and measuring growth and mortality to characterize fitness and overall stress. Once these oysters reached adulthood, we used a quantitative breeding design and reared their offspring at low or ambient salinity in the laboratory. To determine the effects of the larval rearing environment on larval fitness, we measured size and mortality 5 dpf. We focused on larval size, because it is an important determinant of fitness: larger larvae typically have more energy available for metamorphosis, thus reducing their time in the water column when predation is high and giving them a competitive advantage after settlement [37,38]. To test for effects of TGP and potential mechanisms, we

measured the influence of parental acclimation salinity conditions and egg quality on larval size. Finally, we quantified the presence of additive genetic variation for larval body size when reared at either low or ambient salinity to determine the population's capacity to adapt to future changes in salinity.

## 2. Methods

### (a) Population origin and parental acclimation

Parental oysters were produced in summer 2016 by spawning broodstock at the Louisiana Department of Wildlife and Fisheries Michael C. Voisin (MCV) Oyster Hatchery located in Grand Isle, LA (29°14′17.98″ N, 90°00′09.98″ W). Broodstock used in this hatchery were collected in early 2016 by dredging from nearby populations in coastal Louisiana; however, hatchery records on which population was used as broodstock to generate the parents used in our experiment were lost after spawning. Broodstock were maintained in bags on an adjustable longline system at the Louisiana Sea Grant Oyster Research and Demonstration Farm (29° 14′20.3″ N, 90°00′11.55″ W) in Grand Isle prior to spawning. Spawning was induced using warm temperatures (28°C) and the addition of sterilized sperm. Oysters were raised in the hatchery in an upwelling system for four months before being deployed in bags on adjustable longline systems at the low- and medium-salinity acclimation sites in February 2017. A total of 240 oysters were deployed at each acclimation site into three replicate longline bags where they acclimated for 2 years; a low-salinity site at Louisiana Universities Marine Consortium (LUMCON; 29°15′12.5″ N 90°39′46.0″ W) and a medium-salinity site at the research farm in Grand Isle (figure 1a). Mean daily water quality data for our acclimation sites are regularly collected by LUMCON and the USGS site Caminada Pass (29°13′52.9″ N, 90°02′54.7″ W), located approximately 5 miles from our Grand Isle site (figure 1b). These two sites share similar temperature and DO content and primarily differ in salinity (figure 1b).

### (b) Crosses and larval culturing

In May 2019, we collected oysters that had been acclimated at the two salinity sites (LUMCON and Grand Isle) for 2 years. These oysters were brought to the MCV Oyster Hatchery in Grand Isle to perform crosses. Crosses were made following a modified North Carolina II breeding block design (figure 1c) [4]. We performed 16 experimental blocks, where gametes from two males and four females were combined for a total of eight separate fertilizations. Each block consisted of two males (one from each acclimation site) and four females (two from each acclimation site). For each experimental block, eggs from two individual dams from LUMCON were independently crossed with sperm from two individual sires, one from LUMCON and one from Grand Isle. Eggs from two individual dams from Grand Isle were independently crossed with sperm from two individual sires, one from LUMCON and one from Grand Isle. Young oyster cohorts are often skewed male dominant, resulting in uneven sex ratios and an inadequate number of female oysters available for us to spawn. Subsequently, some females were used in multiple blocks and were therefore crossed with a total of four males (electronic supplementary material). A total of 16 males were used from each acclimation site. A total of 19 females were used from LUMCON and 21 females were used from Grand Isle.

Oysters were strip spawned and their sex and gamete quality were assessed under a microscope. Sperm was washed on a 13 μm filter while eggs were washed on a 35 μm filter to remove debris. Fertilization occurred at 15 psu and 28°C at an egg concentration of 10 000 eggs per ml in 100 ml. Due to the difficulty in visually determining egg and sperm quality, many crosses had

poor fertilization success; approximately 68% of the families were discarded due to inadequate larval densities before exposure to treatment conditions (electronic supplementary material).

For crosses that had successful fertilization ($n = 40$), larvae from each family were split into a low (8 psu) and ambient (15 psu) rearing salinity. To minimize experimental treatment shock for the low-salinity treatment, larvae were transferred to a salinity of 11.5 psu 24 h post-fertilization. Following 48 h post-fertilization, larvae were transferred to their final salinity conditions (15 psu or 8 psu). The ambient salinity of 15 psu is typical of the salinities experienced on oyster reefs in coastal Louisiana during the early summer spawning season (USGS Water Data). We chose a low-salinity treatment that was stressful but still within the range that oysters typically experience during spawning season in this region. Algae are grown at the MCV oyster hatchery and larvae were fed in equal ratios of five species: *Tisochrysis lutea, Isochrysis galbana, Pavlova lutheri, Chaetoceros muelleri* and *Chaetoceros calcitrans.* Larvae were fed twice a day at 20 000–25 000 cells ml$^{-1}$ on day 2 and 3 and 30 000–40 000 cells ml$^{-1}$ on day 4 and 5 [39]. Five days post-fertilization, larvae were collected and stained with neutral red dye for 30 min and preserved with buffered formaldehyde added at 5% concentration.

### (c) Survival and morphometrics

We measured mortality and body size on adults that had been outplanted to each of the two field acclimation sites. Mortality was assessed every two months once oysters were large enough to be deployed at the two acclimation sites (February 2017–October 2018). We measured the shell height on 154 surviving oysters that were acclimated at Grand Isle and 126 surviving oysters that were acclimated at LUMCON in May 2019. Using a one-way ANOVA, we tested the effects of acclimation site on adult mortality and size using the lme4 package [40] in R v. 3.4.4. For the larvae, we estimated survival and measured body size on photographs of larvae preserved at five days post-fertilization (dpf) (Nikon Digital Sight DS-Fi2). Survival was estimated by counting the ratio of empty shells to neutral red stained larvae. Using ImageJ under 13.5× compound magnification, we measured body size (maximum anterior–posterior length) for 1248 and 1379 larvae reared at low and ambient salinity, respectively (approx. 50 larvae per cross per treatment). Using a three-way ANOVA, we tested the effects of larval rearing salinity, dam acclimation salinity and sire acclimation salinity on larval body size with Dam ID and Sire ID as random factors in the car package [41]. We confirmed that block ID did not need to be included as a random effect by plotting the residuals of the model against block ID (electronic supplementary material, figure S1). We further tested these effects using a Tukey's 'honest significant difference' method in the package emmeans [42] in R. Despite poor fertilization that limited the number of families available for exposure to experimental treatments, this did not result an unbalanced design. There were 10 dams and 7 sires from each site (LUMCON and GI) that contributed to successful fertilizations that resulted in 34 families at low salinity and 35 families at ambient salinity for statistical analyses.

### (d) Genetic (co)variation and heritability

To determine whether larval size under low or ambient salinity has the capacity to evolve, we estimated variance components under both low and ambient larval rearing salinity. Variance components and heritability were estimated from 12 sires, 21 dams and 27 families at low salinity, and 13 sires, 24 dams and 29 families at ambient salinity (electronic supplementary material). We used an animal model which is a form of a mixed model that can estimate the genetic and environmental components of phenotypic variation using pedigree data [43]. Specifically, we used a generalized linear mixed model (GLMM) using Markov

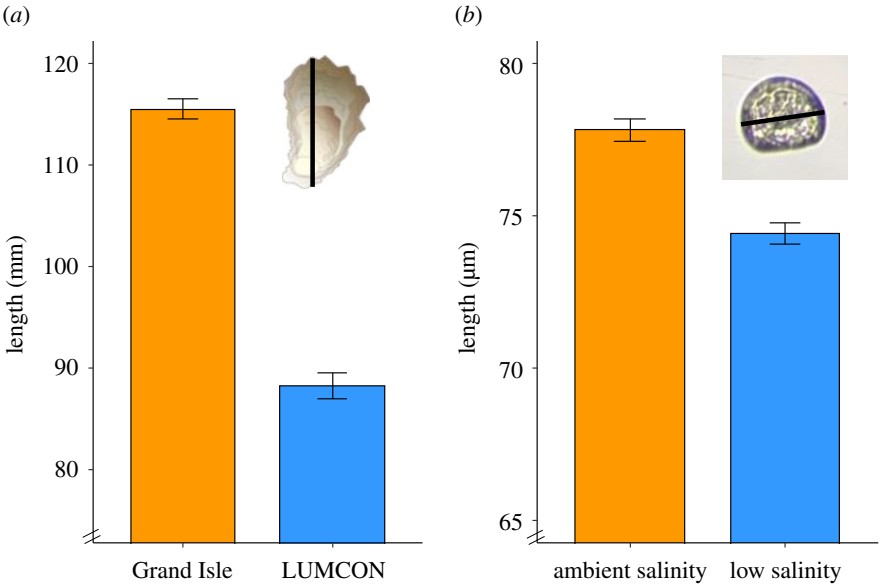

**Figure 2.** (a) Shell height of adult parental oysters reared at Grand Isle (medium salinity) or LUMCON (low salinity) for 2 years. Inset shows linear measurements for adults. (b) Mean anterior–posterior length of offspring (5 dpf) reared at either ambient (15 psu) or low salinity (8 psu). Inset shows *C. virginica* larva at 5 dpf showing linear measurement for max anterior–posterior length at 100× magnification. (Online version in colour.)

chain Monte Carlo (MCMC) in the R package MCMCglmm [44] in R v. 3.4.4. These models are particularly useful for estimating genetic variance in unbalanced pedigrees such as in wild populations or a lack of a fully crossed breeding designs [45]. Following tutorials in Wilson *et al.* [45], we estimated variance components and narrow-sense heritability ($h^2$) for larval size at each larval rearing salinity using the model:

$$y = \mu_+ Z_1 \text{Animal} + Z_2 \text{DamID} + \text{ß}_1 \text{DamAcclim} + \text{ß}_2 \text{SireAcclim} + \varepsilon,$$
(2.1)

where $\mu$ is the population mean, Animal is a random effect of the additive genetic effects of each individual and DamID is a random effect due to non-genetic components (i.e. maternal effects) that also contribute to phenotypic variance. Finally, dam and sire acclimation site (DamAcclim and SireAcclim, respectively) are fixed effects and $\varepsilon$ is a random residual error. We calculated narrow-sense heritability as the ratio of additive genetic variance and total phenotypic variance. We also calculated maternal effects, which is the ratio of variance contributed by each dam ID to the total phenotypic variance. Finally, we ran a similar model to equation (2.1) to estimate the additive genetic covariance for larval size between low- and ambient salinity rearing conditions. For this model, larval length was averaged for each family and Animal in the model is a random effect for each family rather than the individual. After estimating the heritability of larval size under low salinity, we estimated the strength of selection necessary to maintain the same body size under low-salinity conditions as under ambient conditions using the breeder's equation (electronic supplementary material).

The package MCMCglmm uses a Bayesian framework to estimate the variance contributed by each effect in the model. Thus, we set the priors to equally partition variation among all random effects (Animal, DamID and residual error). For the covariance matrix, we used an inverse-Wishart distribution to define priors [46]. We confirmed that other priors did not significantly change the outcome of the model by comparing posterior results. The MCMC chains were run for 1 300 000 generations with a burn-in of 300 000, which we then visually checked for convergence as recommended in the Wilson *et al.* [45] tutorial. Our autocorrelation values for the parameters were near zero, confirming that convergence had occurred and there were no trends in the parameters over successive generations of the model.

## (e) Egg quality assessment

To determine if transgenerational effects may be explained by egg quality differences among dams, we assessed the major biochemical content in oyster eggs for each dam (amounts of total protein and different classes of lipid). Eggs were collected and flash-frozen from females used in crosses that were reared at the low- and medium-salinity site ($n = 15$ from each site). Methods for biochemical measurements can be found in the electronic supplementary material.

Each egg quality measurement (protein, hydrocarbons, triacylglycerols, free fatty acids, sterols and phospholipids) was normalized by a log transformation and then fit with a linear mixed effect model in R. To determine if crosses with 100% mortality were the result of poor egg quality, we ran a one-way ANOVA using the nlme package in R. A two-way ANOVA was performed to test the influence of dam acclimation conditions on egg quality. Finally, the influence of egg quality on larval body size was assessed by running a one-way ANOVA with DamID and SireID as random effects using the lme4 package [40] in R v. 3.4.4.

## 3. Results

## (a) Environmental contributions to adult and larval size

We first tested the influence of acclimation history on adult oyster mortality and size. Cumulative mortality was similar across sites with a mean mortality of 30.2% and 35.0% after 2 years of acclimation at the medium- and low-salinity site, respectively ($F_{1,364} = 0.38$, $p = 0.57$; electronic supplementary material, figure S2). However, we found that the parents were 40% (±0.01% s.e.) larger when reared at the medium-salinity site ($F_{1,278} = 313$, $p < 0.0001$; figure 2a), confirming that the rearing environment has a significant effect on size. We also tested the influence of environmental effects on larval mortality and size by rearing offspring at either low or ambient salinity for 5 days. We observed no differences in mortality between low or ambient larval rearing salinities ($F_{1,28.3} = 0.16$, $p = 0.70$; electronic supplementary material, table S1), with mean mortalities around 27%. For larval

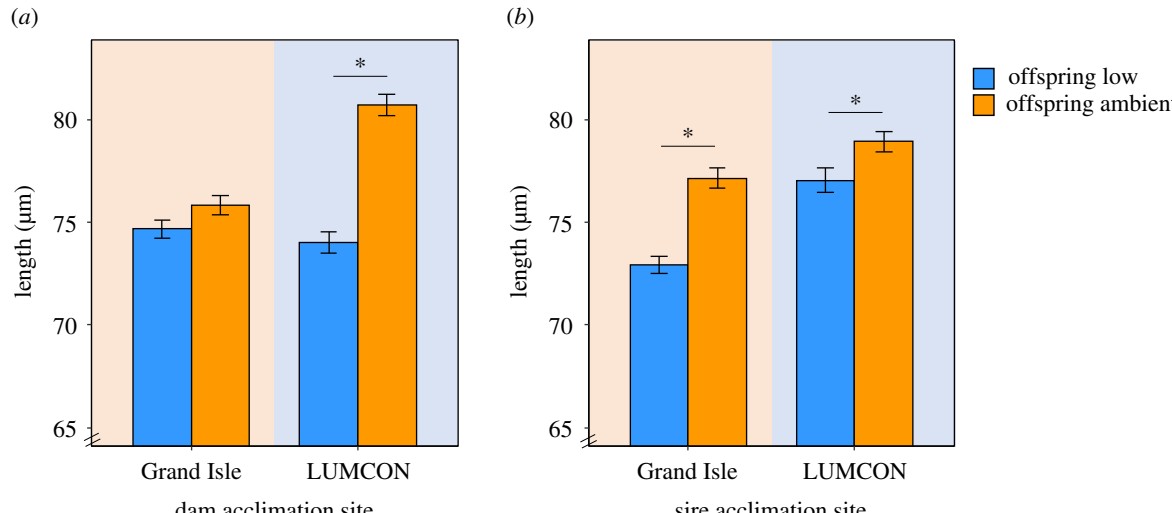

**Figure 3.** Mean anterior–posterior length of larvae reared at either ambient (15 psu) or low salinity (8 psu). Background colours represent larvae with a dam or sire that was acclimated to GI (medium-salinity site in orange) or LUMCON (low-salinity site in blue). (*a*) Comparison of dam acclimation effect on larvae length. (*b*) Comparison of sire acclimation effect on larvae length. Tukey post hoc results are represented as an asterisk over significant comparisons ($p < 0.05$). (Online version in colour.)

size, we found that larvae reared at ambient salinity were 4.5% (±0.46% s.e.) larger ($F_{1,2607} = 39.5$, $p < 0.001$; electronic supplementary material, table S2; figure 2*b*).

### (b) Transgenerational plasticity contributions to larval size

To test for effects of TGP, we compared mortality and size in larvae with parents from different acclimation sites. We observed no differences in larval mortality between dam acclimation site ($F_{1,37} = 0.02$, $p = 0.89$; electronic supplementary material, table S1), nor sire acclimation site ($F_{1,36.8} = 0.64$, $p = 0.43$; electronic supplementary material, table S1). However, TGP effects contributed by dams had a significant effect on larval size (figure 3*a*). When dams were acclimated at the low-salinity site, we observed a significant difference in larval size between low and ambient larval rearing salinities ($F_{1,2613} = 43.7$, $p < 0.001$; electronic supplementary material, table S2; figure 3*a*). Larvae reared in ambient salinity conditions were 9.1% (±0.64% s.e.) larger compared to low-salinity conditions when their dams were acclimated at the low-salinity site (Tukey, $p < 0.05$; figure 3*a*; electronic supplementary material, table S3). There was no difference in size for larvae reared at low and ambient salinities when their dams were acclimated at the medium-salinity site (Tukey, $p > 0.05$; figure 3*a*; electronic supplementary material, table S3).

TGP effects contributed by sires had a nearly significant effect on larval body size ($F_{1,2611} = 3.7$, $p = 0.055$; figure 3*b*; electronic supplementary material, table S2). When sires were acclimated at the medium-salinity site, larvae were 5.8% (±0.67% s.e.) larger when reared at ambient salinity compared to low salinity (Tukey, $p < 0.05$; figure 3*b*; electronic supplementary material, table S3). When sires were acclimated at the low-salinity site, larvae were 2.5% (±0.63% s.e.) larger when reared at ambient salinity compared to low salinity (Tukey, $p < 0.05$; figure 3*b*; electronic supplementary material, table S3).

To further tease apart the mechanism of TGP in dams, we measured egg quality (protein and lipid-class contents) of females. We observed no differences in egg quality between dam acclimation sites ($p > 0.05$; electronic supplementary material, table S4), and egg quality did not influence fertilization success ($p > 0.05$; electronic supplementary material, table S5). Finally, we tested whether egg quality was a good predictor of larval size; however, we observed no correlations among any egg quality measurement and larval size ($p > 0.05$; electronic supplementary material, table S6).

### (c) High genetic (co)variance and capacity to evolve

We estimated the capacity of this population to evolve to either low or ambient salinity in the offspring generation. We found significant and similar levels of additive genetic variance for larval size at low and ambient rearing salinity (table 1; electronic supplementary material, table S7). Consequently, estimates of narrow-sense heritability were similar at ambient ($h^2 = 0.66$) and low ($h^2 = 0.68$) rearing salinity (figure 4; electronic supplementary material, table S7). Nevertheless, these high narrow-sense heritability estimates suggest that there is ample genetic variation present for oysters to adapt to either low or ambient salinity conditions. In addition, mean larval size for each family reared at low and ambient salinity were strongly correlated ($F_{1,26} = 31.9$, $p < 0.0001$; table 1), and the additive genetic covariance for mean larval size for each family between low and ambient rearing salinity was positive and significantly different from zero (table 1; electronic supplementary material, table S8). These relationships suggest that some genotypes perform better than others regardless of the environment.

We were also able to compare the relative contributions of parental genotype and TGP in determining larval size. TGP was measured from maternal effects in our animal model, which had a significantly lower influence on larval size at both ambient (0.16) and low (0.11) larval rearing salinity compared to heritability estimates (figure 4; electronic supplementary material, table S7), again stressing the importance of genotype.

## 4. Discussion

Estuarine salinity conditions in the northern Gulf of Mexico are predicted to undergo drastic changes due to climate

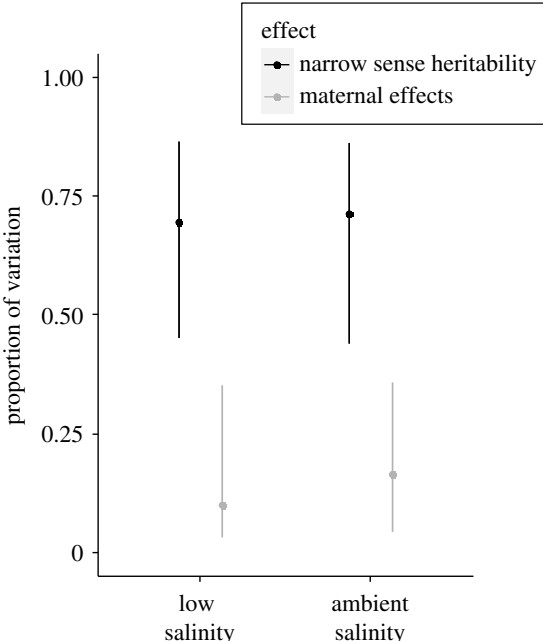

**Figure. 4.** Narrow-sense heritability and maternal effects (with 95% confidence intervals) estimated for larval size reared at either low or ambient salinity.

**Table 1.** Additive genetic variance and covariance for larval size (in µm) from three animal models. Additive genetic variances estimated from individual sizes are displayed on the diagonal for each larval salinity exposure. Additive genetic covariance estimated from mean family sizes are displayed below the diagonal. In parentheses are 95% confidence intervals. Full results for both models can be found in electronic supplementary material, tables S7 and S8. The $r^2$ correlation value for mean family size at low and ambient salinity is displayed above the diagonal.

|        | 8 psu                 | 15 psu              |
|--------|-----------------------|---------------------|
| 8 psu  | 140.7 (68.6–176.0)    | $r^2 = 0.5333$      |
| 15 psu | 55.18 (9.89–110.98)   | 122.3 (65.1–180.1)  |

change and coastal management activities [34,35], which will negatively affect oyster productivity [36]. Our results are in agreement with many other studies that have observed reduced fitness at low salinities at multiple life stages [31,33,47]. We observed reduced growth, but similar mortality, for both adults and larvae under low-salinity conditions. Adult mortality at LUMCON is most likely to be explained by prolonged low-salinity exposure, whereas mortality at GI is most likely to be caused by infection of the parasite *Perkinsus marinus*. While higher salinity conditions are more conducive to increased growth, infection of *P. marinus* intensity is also higher [48]. For larvae, we observed reduced growth under low-salinity conditions. Although these were relatively small reductions in size (4.5%), the carry-over effects from the larval phase may persist and amplify into the juvenile and adult phase. For example, metamorphosis from the larval to juvenile phase is energetically costly, and completing metamorphosis at smaller sizes results in increased mortality and decreased growth in juvenile and adult phases [49,50]. Even if larvae delay metamorphosis and remain in the water column as larvae, this increases predation risk [49]. The size reductions reported here are based on modern populations in future low-salinity conditions

and TGP and evolutionary adaptation may modulate these responses. We demonstrate the importance of standing genetic variation for the persistence of *C. virginica* in future low-salinity conditions.

## (a) Influence of transgenerational plasticity

TGP is predicted to be adaptive when the parental environment matches that of the offspring environment [22]. However, we did not observe increased growth in larvae reared in low-salinity conditions when their dams were acclimated to the low-salinity site. This suggests that TGP contributed by dams may not be a primary mechanism to increase offspring fitness in future low-salinity environments. Not all species seem to benefit from matching stressful parental and offspring conditions, with some species producing poor-quality offspring [26,51]. Reduced body sizes of adults acclimated to the low-salinity site would suggest that these parents may have been under higher levels of stress, resulting in less energy for reproduction. While we observed no differences in egg quality among acclimation sites, the energy budget could be revealed by measuring egg sizes among dams. Thus, the lack of TGP effects may have been overshadowed by variation in initial egg sizes among dams.

Alternatively, we may not have observed an effect of TGP because larvae were not exposed to their final low-salinity condition until two dpf. At this stage, larvae have already developed their shells, which has been shown to be a sensitive developmental marker in larval responses to stressful salinity and pH conditions [52,53]. In a marine tubeworm, TGP has been demonstrated to be more pronounced when both the fertilization and developmental environment of the offspring match that of the parental environment [54].

Nevertheless, we observed that TGP contributed by dams that were acclimated at the low-salinity site was an important contributor to larval size when reared in ambient salinity conditions. Contrary to predictions, our results suggest that a mismatch between the parent and offspring environment can promote offspring growth in this species, and that stressful maternal conditioning may have an adaptive effect that promotes higher fitness when their offspring are reared in a nonstressful environment [25]. This effect may be the result of mothers producing larger, but possibly fewer eggs, when they are in a low-quality or low food availability environment [55,56]. While we did not estimate egg counts or egg sizes among females, we were able to demonstrate that there was no difference in dams' egg quality between the medium and low acclimation sites (electronic supplementary material, table S4). This suggests that maternal effects transferred through egg quality may not have played a role in the TGP effects we observed. In addition, previous studies have observed that higher egg quality is not always correlated with faster larval growth [57,58]. These results suggest that other mechanisms of maternal effects, besides egg quality, are important for larval fitness, such as egg quantity and size or epigenetics [20]. High epigenetic divergence has previously been demonstrated among populations of *C. virginica* in Louisiana, potentially in response to salinity gradients in the Gulf of Mexico [59].

We also observed higher growth in larvae reared in ambient salinity conditions when sires were acclimated at the low-salinity site, which may be explained by epigenetic effects. Paternal effects have been observed in other species

[60], with salinity specific examples in marine tubeworms and oysters [54,61]. These observations of paternal TGP in multiple marine systems highlights the need for additional research into the role of paternal effects in transgenerational epigenetic inheritance.

## (b) High capacity to evolve

Typically, highly plastic traits have low genetic variation and heritability estimates [4,62], since the trait is largely controlled by the environment. Despite the high plasticity we observed in body size, narrow-sense heritability estimates were high, suggesting that there is ample genetic variation present within the population to adapt to both low- and ambient salinity conditions. The relative contribution of parental genotype (heritability) was higher than maternal effects (TGP and epigenetics). This suggests that evolution could be a primary mechanism through which populations of *C. virginica* may persist in future low-salinity conditions. While other studies observe either reduced genetic variation [10,63], or cryptic (increased) genetic variation [64] under stressful conditions, we find similar levels of genetic variation in larvae reared in ambient and low-salinity conditions. Our low-salinity conditions may not have been stressful enough to reveal either reduced or cryptic genetic variation. Another study observed lower narrow-sense heritability estimates when adult *C. virginica* were exposed to 3 psu [65]. Our results may also be explained by the positive genetic covariance detected between larval size at low and ambient salinity rearing conditions. Families that have high growth rates at ambient salinity also have higher growth rates at low-salinity conditions, suggesting that there is no trade-off for growth at stressful low-salinity conditions. Given that salinity fluctuations in the estuary occur on a daily and seasonal basis, we expect evolutionary adaptation for growth to occur more quickly than in the absence of covariation.

For a population that was exposed to low salinity (with a resulting mean reduction in the size of 3.4 µm as measured in this experiment) and a heritability of 0.68, the calculated selection differential necessary to maintain the same size as under ambient salinity is 5 µm generation$^{-1}$ (electronic supplementary material). The percentage mortality required to achieve this selection differential in just one generation would be just 25% (i.e. mortality of the smallest quartile of larvae); a relatively small reduction in population size that we would expect to have minimal long-term impacts for population recovery to low-salinity exposure.

## 5. Conclusion and future implications

We investigated the mechanism of response to low salinity in *C. virginica* and estimated the adaptive capacity of larval size to evolve. We found that variance in larval size was primarily controlled by parental genotype, suggesting that *C. virginica* may be able to adapt to future anthropogenic changes in salinity. These results suggest that selective breeding in hatchery management practices may be an effective way of increasing resiliency for low-salinity tolerance in *C. virginica*. However, low-salinity conditions in the wild coincide with stressful temperature spikes in the summer months, thus we expect oysters to experience these stressors in tandem. The presence of genetic variation for thermal tolerance in *C. virginica* is less well known [66], including the combined effects of thermal and salinity stress on genetic variation for tolerance. Current research has documented reduced low-salinity tolerance at warmer temperatures [33], which may translate to reduced heritability for combined low-salinity and high-temperature exposure. Future research should focus on combined stressors recorded in the field to determine net effects on genetic variation and heritability estimates.

Data accessibility. Raw data (size and mortality datasets) and scripts are available from the Dryad Digital Repository: https://doi.org/10.25338/B8790R [67] and GitHub https://github.com/JoannaGriffiths/Cvirginica_TGP_Heritability [68]. The data are provided in the electronic supplementary material [69].

Authors' contributions. J.S.G.: conceptualization, data curation, formal analysis, investigation, methodology, project administration, writing-original draft, writing-review and editing; K.M.J.: data curation, methodology, writing-review and editing; K.A.S.: data curation; M.S.Y.: data curation; F.T.C.P.: data curation, formal analysis, methodology, resources and writing-original draft; J.F.L.P.: data curation, resources, writing-review and editing; M.W.K.: conceptualization, data curation, funding acquisition, methodology, resources, supervision, writing-review and editing. All authors gave final approval for publication and agreed to be held accountable for the work performed therein.

Funding. This work was supported by a Louisiana Environmental Education Committee (LEEC) grant awarded to J.S.G.; a NSF-BioOCE 1731710 award to M.W.K. and J.F.L.; and a Louisiana Sea Grant award NA14OAR4170099 to M.W.K. and J.F.L.P.

Competing interests. We declare we have no competing interests.

Acknowledgements. We would like to thank Dr Brian Callam and the Grand Isle Oyster Hatchery with help performing crosses and providing space for experiments. We also thank Dr Donal Manahan (USC) for use of his laboratory for protein and lipid analyses. We thank the two anonymous reviewers for feedback that improved this manuscript.

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
