## [Peer Review File · Proceedings of the Royal Society B: Biological Sciences]

Review History

RSPB-2020-3118.R0 (Original submission)

Review form: Reviewer 1

Recommendation

Major revision is needed (please make suggestions in comments)

Scientific importance: Is the manuscript an original and important contribution to its field?

Good

General interest: Is the paper of sufficient general interest?

Good

Quality of the paper: Is the overall quality of the paper suitable?

Acceptable

Is the length of the paper justified?

Yes

Should the paper be seen by a specialist statistical reviewer?

No

Do you have any concerns about statistical analyses in this paper? If so, please specify them explicitly in your report.

No

It is a condition of publication that authors make their supporting data, code and materials available - either as supplementary material or hosted in an external repository. Please rate, if applicable, the supporting data on the following criteria.

Is it accessible?

Yes

Is it clear?

Yes

Is it adequate?

Yes

Do you have any ethical concerns with this paper?

No

Comments to the Author

The study by Griffiths et al assessed the larvae of acclimated adult oysters in control and low salinity. They find that when parents are reared in low salinity, larvae still show decreased size, i.e. there were no positive transgenerational effects. However, they find heritable genetic variance for body size in low salinity where parental genotype is important in determining larval success. The contrast of transgenerational impacts versus genetic variation is very interesting and especially important to determine for this economically important species in a changing ocean. The manuscript is well organised and nicely written however more information in the introduction is needed to provide important background information related to the experiments, as well as further details on the complex experimental design.

Abstract

Line 60: how long did you raise larvae under low or high salinity conditions?

Introduction

The introduction was well written and flowed well. More information, however, is needed about impacts of egg quality on larval performance since this was a main part of your methods. E.g. a recent paper by Swezey et al., 2020 in PNAS would be very appropriate here (doi.org/10.1073/pnas.2006910117).

I would recommend outlining the aims of the experiment at the end so that the sections in the methods are intuitive, for example, I did not expect an analysis of survival and morphometrics of oysters that had been outplanted, only an analysis of their larvae. There are a lot of interesting aspects to the experiments performed and would be great to see them laid out here clearly.

Line 80: Could you briefly summarise some of these studies?

Methods

I could not follow well how many sires and dams were used and how many resultant crosses there were throughout the methods. Were any of the crosses replicated? Did you only use one of each parent from an acclimated salinity, or did you have crosses where both parents were from the same salinity?

Lines 183-190: This is a little confusing – how many individual male x female crosses did you have at the end? Why did you use multiple females per male? Were blocks all completed at the same time? I see in the supplementary table that it not a straightforward crossing of males and females – I think a more detailed figure could help this, also indicating what the different “blocks” mean. (Also should this be indicated in the text as Fig. 1C not 1B on line 183?)

Please indicate the version of R you used for the analysis.

Line 193: how long after fertilisation were embryos transferred to salinity treatments?

Lines 210-214: is this common with raising oysters to have such high mortality – and do you expect this initial selection to skew your analyses, ie. did this create uneven sire/dam IDs across the treatments? Did some genotypes perform badly across both treatments because they were a bad genotype, or was it because of mortality from the treatment?

From the original number of crosses, sires and dams – how many survived across treatments?

Could you specify the analysis performed with the ones that survived – currently you refer to it as “the following analyses” but directly below the egg quality assessment was done with a different set of females.

Section – Egg Quality Assessment: how many females from each salinity were assessed?

Line 252: how many oysters were outplanted to the field acclimation sites? How many replicates per salinity were there?

Results

I like the clear layout of the results, it was easy to follow.

15psu is the “control” or ambient treatment? If so, I would recommend referring to this as the control, rather than “high” salinity site throughout the manuscript.

Line 321-322: how did you assess this? Is this based on size, or larval stage – which would good way to assess this.

Line 310: in the methods it is mentioned that mortality was measured every 2 months, what did these results look like? Was mortality even along time for both treatments?

Discussion

Could you please include some discussion on mortality of the adult oysters across the treatments?

I also suggest citing the review by Byrne et al. 2020 (10.1111/gcb.14882) which addresses similar themes in your manuscript.

These are quite small changes (<5%) in body size between treatments, how much do you expect this to impact settlement and survival? I think a greater discussion on how size impacts survival is needed as this was a main component of your experiments.

Lines 452-454: “Egg quality had no influence on larval growth” is quite a strong conclusion, especially as fertilisation was not performed at the experimental low salinity, where different impacts of dams could have been revealed here.

I think that a discussion about how increased temperature might affect your results is important, especially as previous studies have examined multistressor impacts. Do you expect oysters to be able to adapt to both stressors?

Would you be able to put the narrow sense heritability numbers into context? How quickly would you expect these animals to adapt with these kind of values?

Figures

Figure 1C – I would make this a separate figure with much more details as described above.

Figure 4: could you use a lighter grey to distinguish from the black? I found it difficult to tell the difference between narrow sense and maternal effects in the figure.

Review form: Reviewer 2

Recommendation

Accept with minor revision (please list in comments)

Scientific importance: Is the manuscript an original and important contribution to its field?

Excellent

General interest: Is the paper of sufficient general interest?

Good

Quality of the paper: Is the overall quality of the paper suitable?

Excellent

Is the length of the paper justified?

Yes

Should the paper be seen by a specialist statistical reviewer?

No

Do you have any concerns about statistical analyses in this paper? If so, please specify them explicitly in your report.

No

It is a condition of publication that authors make their supporting data, code and materials available - either as supplementary material or hosted in an external repository. Please rate, if applicable, the supporting data on the following criteria.

Is it accessible?

No

Is it clear?

No

Is it adequate?

No

Do you have any ethical concerns with this paper?

No

Comments to the Author

I carefully reviewed this entire paper. The rationale for the study, methods, data, and results were all very clearly presented with a very nice discussion. I complement the authors on such a clear presentation. I think this is an important paper to publish because the authors do not find transgenerational effects of adult exposure to salinity on larvae. Publishing these results helps to avoid publication bias, and it also relevant to discussions in aquaculture on how adults should be conditioned to maximize survival of larvae. I have only a few minor comments.

- statistical model - Why wasn't Block ID included? Could you plot the residuals of the model against Block ID to check that this should not be included as a random effect? (e.g. if the residuals do not show the same patterns across blocks, then it should be included)

- Was egg size measured? It seems to me that there may be transgenerational effects on growth, but these would be overshadowed by the large variability in the egg/larvae size from different dams. Controlling for initial egg size may reveal these. If it was not measured, this point should be added to the discussion.

- Were the adults strip spawned? (line 191) Or induced with temperature and sperm? I would worry a bit if it was the latter, as the sperm could have contaminated the eggs.

- The data accessibility is poor. The github link does not work. The Dryad link works, but it is a bunch of data files (with no metadata) and scripts. There is no README.

Decision letter (RSPB-2020-3118.R0)

25-Feb-2021

Dear Dr Griffiths:

Your manuscript has now been peer reviewed and the reviews have been assessed by an Associate Editor. The reviewers' comments (not including confidential comments to the Editor) and the comments from the Associate Editor are included at the end of this email for your reference. As you will see, the reviewers and the Editors have raised some concerns with your manuscript and we would like to invite you to revise your manuscript to address them.

Research ethics:

Use of animals and field studies:

It is a condition of publication that you make available the data and research materials supporting the results in the article. Please see our Data Sharing Policies (<https://royalsociety.org/journals/authors/author-guidelines/#data>). Datasets should be deposited in an appropriate publicly available repository and details of the associated accession number, link or DOI to the datasets must be included in the Data Accessibility section of the

article (<https://royalsociety.org/journals/ethics-policies/data-sharing-mining/>). Reference(s) to datasets should also be included in the reference list of the article with DOIs (where available).

Please submit a copy of your revised paper within three weeks. If we do not hear from you within this time your manuscript will be rejected. If you are unable to meet this deadline please let us know as soon as possible, as we may be able to grant a short extension.

Best wishes,
Dr Daniel Costa
mailto: proceedingsb@royalsociety.org

Reviewer(s)' Comments to Author:

Referee: 1

Comments to the Author(s)

The study by Griffiths et al assessed the larvae of acclimated adult oysters in control and low salinity. They find that when parents are reared in low salinity, larvae still show decreased size, i.e. there were no positive transgenerational effects. However, they find heritable genetic variance for body size in low salinity where parental genotype is important in determining larval success. The contrast of transgenerational impacts versus genetic variation is very interesting and especially important to determine for this economically important species in a changing ocean. The manuscript is well organised and nicely written however more information in the introduction is needed to provide important background information related to the experiments, as well as further details on the complex experimental design.

Abstract

Line 60: how long did you raise larvae under low or high salinity conditions?

Introduction

The introduction was well written and flowed well. More information, however, is needed about impacts of egg quality on larval performance since this was a main part of your methods. E.g. a recent paper by Swezey et al., 2020 in PNAS would be very appropriate here (doi.org/10.1073/pnas.2006910117).

I would recommend outlining the aims of the experiment at the end so that the sections in the methods are intuitive, for example, I did not expect an analysis of survival and morphometrics of oysters that had been outplanted, only an analysis of their larvae. There are a lot of interesting aspects to the experiments performed and would be great to see them laid out here clearly.

Line 80: Could you briefly summarise some of these studies?

Methods

I could not follow well how many sires and dams were used and how many resultant crosses there were throughout the methods. Were any of the crosses replicated? Did you only use one of each parent from an acclimated salinity, or did you have crosses where both parents were from the same salinity?

Lines 183-190: This is a little confusing – how many individual male x female crosses did you have at the end? Why did you use multiple females per male? Were blocks all completed at the same time? I see in the supplementary table that it not a straightforward crossing of males and females – I think a more detailed figure could help this, also indicating what the different “blocks” mean. (Also should this be indicated in the text as Fig. 1C not 1B on line 183?)

Please indicate the version of R you used for the analysis.

Line 193: how long after fertilisation were embryos transferred to salinity treatments?

Lines 210-214: is this common with raising oysters to have such high mortality – and do you expect this initial selection to skew your analyses, ie. did this create uneven sire/dam IDs across the treatments? Did some genotypes perform badly across both treatments because they were a bad genotype, or was it because of mortality from the treatment?

From the original number of crosses, sires and dams – how many survived across treatments?

Could you specify the analysis performed with the ones that survived – currently you refer to it as “the following analyses” but directly below the egg quality assessment was done with a different set of females.

Section – Egg Quality Assessment: how many females from each salinity were assessed?

Line 252: how many oysters were outplanted to the field acclimation sites? How many replicates per salinity were there?

Results

I like the clear layout of the results, it was easy to follow.

15psu is the “control” or ambient treatment? If so, I would recommend referring to this as the control, rather than “high” salinity site throughout the manuscript.

Line 321-322: how did you assess this? Is this based on size, or larval stage – which would good way to assess this.

Line 310: in the methods it is mentioned that mortality was measured every 2 months, what did these results look like? Was mortality even along time for both treatments?

Discussion

Could you please include some discussion on mortality of the adult oysters across the treatments?

I also suggest citing the review by Byrne et al. 2020 (10.1111/gcb.14882) which addresses similar themes in your manuscript.

These are quite small changes (<5%) in body size between treatments, how much do you expect this to impact settlement and survival? I think a greater discussion on how size impacts survival is needed as this was a main component of your experiments.

Lines 452-454: "Egg quality had no influence on larval growth" is quite a strong conclusion, especially as fertilisation was not performed at the experimental low salinity, where different impacts of dams could have been revealed here.

I think that a discussion about how increased temperature might affect your results is important, especially as previous studies have examined multistressor impacts. Do you expect oysters to be able to adapt to both stressors?

Would you be able to put the narrow sense heritability numbers into context? How quickly would you expect these animals to adapt with these kind of values?

Figures

Figure 1C - I would make this a separate figure with much more details as described above.

Figure 4: could you use a lighter grey to distinguish from the black? I found it difficult to tell the difference between narrow sense and maternal effects in the figure.

Referee: 2

Comments to the Author(s)

I carefully reviewed this entire paper. The rationale for the study, methods, data, and results were all very clearly presented with a very nice discussion. I complement the authors on such a clear presentation. I think this is an important paper to publish because the authors do not find transgenerational effects of adult exposure to salinity on larvae. Publishing these results helps to avoid publication bias, and it also relevant to discussions in aquaculture on how adults should be conditioned to maximize survival of larvae. I have only a few minor comments.

- statistical model - Why wasn't Block ID included? Could you plot the residuals of the model against Block ID to check that this should not be included as a random effect? (e.g. if the residuals do not show the same patterns across blocks, then it should be included)

- Was egg size measured? It seems to me that there may be transgenerational effects on growth, but these would be overshadowed by the large variability in the egg/larvae size from different dams. Controlling for initial egg size may reveal these. If it was not measured, this point should be added to the discussion.

- Were the adults strip spawned? (line 191) Or induced with temperature and sperm? I would worry a bit if it was the latter, as the sperm could have contaminated the eggs.

- The data accessibility is poor. The github link does not work. The Dryad link works, but it is a bunch of data files (with no metadata) and scripts. There is no README.

Author's Response to Decision Letter for (RSPB-2020-3118.R0)

See Appendix A.

Decision letter (RSPB-2020-3118.R1)

26-Apr-2021

Dear Dr Griffiths

I am pleased to inform you that your manuscript RSPB-2020-3118.R1 entitled "Transgenerational plasticity and the capacity to adapt to low salinity in the eastern oyster, *Crassostrea virginica*" has been accepted for publication in Proceedings B.

There are no comments from reviewers but please ensure that you meet all the journal requirements listed below.

- 4) A media summary: a short non-technical summary (up to 100 words) of the key findings/importance of your manuscript.
- 5) Data accessibility section and data citation
Please include a DOI for your data in Dryad and ensure your data is included in the reference list.
- 6) For more information on our Licence to Publish, Open Access, Cover images and Media summaries, please visit <https://royalsociety.org/journals/authors/author-guidelines/>.

Sincerely,
Editor, Proceedings B
<mailto:proceedingsb@royalsociety.org>

Associate Editor:

Board Member

Comments to Author:

The authors did a great job addressing all the reviewers comments. I am please to recommend this for publication in the Evolving Seas Special Feature.

Decision letter (RSPB-2020-3118.R2)

29-Apr-2021

Dear Dr Griffiths

I am pleased to inform you that your manuscript entitled "Transgenerational plasticity and the capacity to adapt to low salinity in the eastern oyster, *Crassostrea virginica*" has been accepted for publication in Proceedings B.

Data Accessibility section

Open Access

Paper charges

Sincerely,
Editor, Proceedings B
mailto: proceedingsb@royalsociety.org

Appendix A

Thank you to both referee's for their incredibly helpful feedback to improve this manuscript. Please see our responses below in blue. We have included a version of this manuscript with tracked changes at the end of this response.

Referee: 1

Comments to the Author(s)

The study by Griffiths et al assessed the larvae of acclimated adult oysters in control and low salinity. They find that when parents are reared in low salinity, larvae still show decreased size, i.e. there were no positive transgenerational effects. However, they find heritable genetic variance for body size in low salinity where parental genotype is important in determining larval success. The contrast of transgenerational impacts versus genetic variation is very interesting and especially important to determine for this economically important species in a changing ocean. The manuscript is well organised and nicely written however more information in the introduction is needed to provide important background information related to the experiments, as well as further details on the complex experimental design.

Abstract

Line 60: how long did you raise larvae under low or high salinity conditions?

-> Larvae were reared until 5 days post-fertilization under treatment conditions. This information was added to the abstract on L57.

Introduction

The introduction was well written and flowed well. More information, however, is needed about impacts of egg quality on larval performance since this was a main part of your methods. E.g. a recent paper by Swezey et al., 2020 in PNAS would be very appropriate here (doi.org/10.1073/pnas.2006910117).

>We included some discussion on this in the introduction (L94-95), but we refrain from going into too much detail since we did not find egg quality to determine the fitness of larvae in our results and we are limited in space. We hope this addition is sufficient.

I would recommend outlining the aims of the experiment at the end so that the sections in the methods are intuitive, for example, I did not expect an analysis of survival and morphometrics of oysters that had been outplanted, only an analysis of their larvae. There are a lot of interesting aspects to the experiments performed and would be great to see them laid out here clearly.

>Thank you for the suggestion, we have now provided a clear outline of our aims that follows the progression of our methods as well as data in our figures (L132-145).

Line 80: Could you briefly summarise some of these studies?

>We provided a brief summary on L75-76.

Methods

I could not follow well how many sires and dams were used and how many resultant crosses there were throughout the methods. Were any of the crosses replicated? Did you only use one

of each parent from an acclimated salinity, or did you have crosses where both parents were from the same salinity?

>We did not replicate crosses. We crossed one male from the low acclimation site with two females from the low acclimation site and two females from the high acclimation site. Also, within the same block we crossed one male from the high acclimation site with two females from the low acclimation site and two females from the high acclimation site. We hope our changes to the methods on L174-181 make this clearer.

Lines 183-190: This is a little confusing – how many individual male x female crosses did you have at the end? Why did you use multiple females per male? Were blocks all completed at the same time? I see in the supplementary table that it not a straightforward crossing of males and females – I think a more detailed figure could help this, also indicating what the different “blocks” mean.

>We used multiple females per male to maximize our power when modeling maternal effects in our heritability model so we could accurately assess the relative contributions of additive genetic variation and maternal effects on larval size. In young oyster cohorts, there are usually more males than females present, thus resulting in an inadequate number of females for us to use in our crosses. This is why some females were required to be used across multiple blocks. We hope our revised explanation on L181-185 clear up some of the confusion. We also added a little bit more detail to the block design in figure 1C, but we prefer to keep it as part of Figure 1 rather than its own separate figure. We are unsure as to how to make the crosses clearer, but we hope that the caption and methods section now adequately addresses the confusion. We made a new supplementary file with figures similar to Fig 1C for all blocks with oyster IDs in each block. We also put a big red X over crosses that had 100% mortality. Blocks were completed over a continuous timespan of three days. We show that Block ID had no effect on larval size (Fig S1 and L221-221).

(Also should this be indicated in the text as Fig. 1C not 1B on line 183?)

> The correct Figure is now referenced (Fig. 1C) on L174.

Please indicate the version of R you used for the analysis.

The R version used for ANOVA and glmmMCMC analyses are indicated on L213 and 237.

Line 193: how long after fertilisation were embryos transferred to salinity treatments?

> On L193-195, we indicate that for the low salinity treatment, larvae were transferred to 11.5 psu 24 hours post-fertilization, and then to 8 psu at 48 hours post-fertilization. We have re-worded this section to ameliorate confusion.

Lines 210-214: is this common with raising oysters to have such high mortality – and do you expect this initial selection to skew your analyses, ie. did this create uneven sire/dam IDs across the treatments? Did some genotypes perform badly across both treatments because they were a bad genotype, or was it because of mortality from the treatment?

>It is common to see this level of unsuccessful fertilization events since egg viability cannot be visually determined. In addition, sperm mobility quickly declines within the hour after sperm is in contact with seawater, making factorial crosses extremely tricky. Crosses with poor fertilization were never exposed to low salinity, therefore, there was no initial selection. In addition, unsuccessful fertilizations were similar whether parents were acclimated to Grand Isle or LUMCON. We had exactly 10 females from LUMCON and 10 females from GI contribute to surviving families and we had exactly 7 males from LUMCON and 7 males from GI contribute to surviving families (L223-227). In addition, we had a similar number of larval families exposed to low salinity (34 families) and ambient salinity (35 families). We found genetic co-variances to be significant, which suggests that some genotypes that performed poorly at low salinity also performed poorly at ambient salinity (as measured by larval size).

From the original number of crosses, sires and dams – how many survived across treatments? Could you specify the analysis performed with the ones that survived – currently you refer to it as “the following analyses” but directly below the egg quality assessment was done with a different set of females.

>Most crosses survived across treatments. We have changed the wording here to better reflect that what we observed was actually poor fertilization success for some crosses before they were even exposed to their final salinity treatments (L189-191).

We rearranged the methods so that the transgenerational and heritability analyses follow the spawning and breeding design section. We also specify the number of dams, sires, and families at low and high salinity in the survival and morphometrics section where these numbers are relevant (L223-227).

Section – Egg Quality Assessment: how many females from each salinity were assessed?

> We assessed egg quality from 15 females from each site, which is now indicated on L269.

Line 252: how many oysters were outplanted to the field acclimation sites? How many replicates per salinity were there?

>We refer to the methods section where we detail the field acclimation for the parents on L160-164. There were 240 oysters in three replicate longline bags that were outplanted at each site.

Results

I like the clear layout of the results, it was easy to follow.

15psu is the “control” or ambient treatment? If so, I would recommend referring to this as the control, rather than “high” salinity site throughout the manuscript.

> We changed our wording to refer to any of the “high” salinity sites or treatment conditions to be referred to as ambient salinity for larval treatment and medium salinity for the Grand Isle acclimation site. We refrain from using the word control since the acclimation sites cannot necessarily be considered a “control” treatment.

Line 321-322: how did you assess this? Is this based on size, or larval stage – which would good way to assess this.

>We omitted this sentence since we did not specifically measure growth rates in larvae (we only measured larval size at a single time point).

Line 310: in the methods it is mentioned that mortality was measured every 2 months, what did these results look like? Was mortality even along time for both treatments?

>We included a supplementary figure that depicts mortality that was measured every 2 months (Figure S2). We show that mortality is cumulative and even along time for both outplant sites (L283-285).

Discussion

Could you please include some discussion on mortality of the adult oysters across the treatments?

>Thank you for the suggestion, we have now included a discussion on adult and larval mortality at the beginning of the discussion (L342-347).

I also suggest citing the review by Byrne et al. 2020 (10.1111/gcb.14882) which addresses similar themes in your manuscript.

>We enjoyed reading the review and thought it was an interesting summary of the field and provided helpful insight for interpreting our results. We cited this review on L364.

These are quite small changes (<5%) in body size between treatments, how much do you expect this to impact settlement and survival? I think a greater discussion on how size impacts survival is needed as this was a main component of your experiments.

>We discuss the impacts of body size in the first paragraph of the discussion (L347-353). We also link changes in size to our heritability estimates and the evolutionary significance (L418-424).

Lines 452-454: "Egg quality had no influence on larval growth" is quite a strong conclusion, especially as fertilisation was not performed at the experimental low salinity, where different impacts of dams could have been revealed here.

>We agree that this statement is too strong, especially since we could not measure the impacts of dam acclimation on fertilization. We have toned down this language and instead suggest that further work is needed to investigate the mechanism of this response (L386).

I think that a discussion about how increased temperature might affect your results is important, especially as previous studies have examined multistressor impacts. Do you expect oysters to be able to adapt to both stressors?

>This is an important consideration since oysters usually experience low salinity during warm summer months. We have included a discussion of potential impacts from multiple stressors in the conclusion (L433-440).

Would you be able to put the narrow sense heritability numbers into context? How quickly would you expect these animals to adapt with these kind of values?

>We have put these estimates into context using the breeder's equation that would predict the percent mortality for just a single generation to restore larval size to ambient salinity conditions. We found the required mortality to be relatively low (25%), suggesting that selection to low salinity may not result in a population crash (L418-424 and ESM File).

Figures

Figure 1C – I would make this a separate figure with much more details as described above.

Maybe change Fig 1C to Grand isle and LUMCON instead of low and high

>Thank you for the suggestion, we have changed Fig1C to Grand Isle and LUMCON instead of low and high. However, we decided to keep this figure as a panel in Figure 1, rather than its own figure. We would like to keep fig1C simple and instead we provide further detail written in the caption and in the methods section (please see comments above).

Figure 4: could you use a lighter grey to distinguish from the black? I found it difficult to tell the difference between narrow sense and maternal effects in the figure.

>We used a lighter grey for maternal effects in figure 4.

Referee: 2

Comments to the Author(s)

I carefully reviewed this entire paper. The rationale for the study, methods, data, and results were all very clearly presented with a very nice discussion. I complement the authors on such a clear presentation. I think this is an important paper to publish because the authors do not find transgenerational effects of adult exposure to salinity on larvae. Publishing these results helps to avoid publication bias, and it also relevant to discussions in aquaculture on how adults should be conditioned to maximize survival of larvae. I have only a few minor comments.

- statistical model - Why wasn't Block ID included? Could you plot the residuals of the model against Block ID to check that this should not be included as a random effect? (e.g. if the residuals do not show the same patterns across blocks, then it should be included)

>Block ID was not originally included as a random effect since some females were used in more than one block (this was because we were unable to find any more females). We plotted the residuals of the model against Block ID and confirmed that residuals showed the same patterns across blocks and does not need to be included as a random effect in our model. We state this in the methods (L220-222 and Fig. S1).

- Was egg size measured? It seems to me that there may be transgenerational effects on growth, but these would be overshadowed by the large variability in the egg/larvae size from

different dams. Controlling for initial egg size may reveal these. If it was not measured, this point should be added to the discussion.

>Unfortunately, we did not measure egg size since we did not take the correct measures to preserve egg structure during flash freezing. We added a section to our discussion about how transgenerational effects may be overshadowed by variation in egg sizes (L364-369).

- Were the adults strip spawned? (line 191) Or induced with temperature and sperm? I would worry a bit if it was the latter, as the sperm could have contaminated the eggs.

>The oysters were strip spawned. This was stated in the previous paragraph, but we have re-organized the spawning methods so that this information flows better (L186-187).

- The data accessibility is poor. The github link does not work. The Dryad link works, but it is a bunch of data files (with no metadata) and scripts. There is no README.

>We have added a README file to the Dryad link. Unfortunately, I am unable to diagnose the issue with the GitHub link—it seems to be working on my end. However, the Dryad link is now an exact replica of the GitHub link. I've pasted the link to GitHub here. If it continues to not work, you can also access it by going to <https://github.com/JoannaGriffiths> -> Repositories -> Cvirginica_TGP_Heritability

Full link: <https://github.com/>

JoannaGriffiths/Cvirginica_TGP_Heritability